# CAR-NK Cells: A Chimeric Hope or a Promising Therapy?

**DOI:** 10.3390/cancers14153839

**Published:** 2022-08-08

**Authors:** Mohamad Sabbah, Ludovic Jondreville, Claire Lacan, Francoise Norol, Vincent Vieillard, Damien Roos-Weil, Stéphanie Nguyen

**Affiliations:** 1Hematology Department, Pitie-Salpetriere Hospital, 75013 Paris, France; 2Centre d’Immunologie et des Maladies Infectieuses (CIMI-Paris), Sorbonne Université, Inserm U1135, CNRS ERL 8255, 75013 Paris, France

**Keywords:** natural killer cells, CAR-NK, immunotherapy, leukemia, lymphoma, allogenic CAR, CAR-T cells

## Abstract

**Simple Summary:**

In recent years, innovative immunotherapy-based treatments have paved the way for a new approach to hematological malignancies. Instead of conventional chemotherapy, T cells have been genetically engineered to detect—and engage their cytotoxicity against—tumor cells, and their success story is astonishing. However, many setbacks—including insufficient efficacy, deficient autologous source, heavy side effects, and a hefty price—limit their use. A promising alternative could be chimeric antigen receptor NK cells, which possess interesting cytotoxicity and minimal graft-versus-host disease risk. In this article, we review the possible sources, the development techniques, the potential advantages, and the challenges faced in the field of chimeric antigen receptor NK cells.

**Abstract:**

Immunotherapy with chimeric antigen receptor-engineered T cells (CAR-T) has revolutionized the treatment landscape of relapsed/refractory B-cell malignancies. Nonetheless, the use of autologous T cells has certain limitations, including the variable quality and quantity of collected effector T cells, extended time of cell processing, limited number of available CAR cells, toxicities, and a high cost. Thanks to their powerful cytotoxic capabilities, with proven antitumor effects in both haploidentical hematopoietic stem cell transplantation and adoptive cell therapy against solid tumors and hematological malignancies, Natural Killer cells could be a promising alternative. Different sources of NK cells can be used, including cellular lines, cord blood, peripheral blood, and induced pluripotent stem cells. Their biggest advantage is the possibility of using them in an allogeneic context without major toxic side effects. However, the majority of the reports on CAR-NK cells concern preclinical or early clinical trials. Indeed, NK cells might be more difficult to engineer, and the optimization and standardization of expansion and transfection protocols need to be defined. Furthermore, their short persistence after infusion is also a major setback. However, with recent advances in manufacturing engineered CAR-NK cells exploiting their cytolytic capacities, antibody-dependent cellular cytotoxicity (ADCC), and cytokine production, “off-the-shelf” allogeneic CAR-NK cells can provide a great potential in cancer treatments.

## 1. NK Cells’ Biology: A Plenitude of Receptors and a Balance between Immunological Self-Tolerance and Immune Surveillance

Natural killer (NK) cells are specific type of lymphocyte, considered to be part of innate immunity, and representing 10 to 15% of total lymphocyte count. They play a crucial role in immune surveillance of malignant cells and virus-infected cells [1]. These natural-born killers play a role defined by their function: elimination of the cells that have lost the expression of one or more molecules of the class I major histocompatibility complex (MHC) on their surface, without prior contact or immunization, and in an antigen-independent manner—unlike B or T cells. Loss of expression of MHC class I molecules on malignant or infected cell surfaces is a well-known escape mechanism for T-lymphocyte immune surveillance, but renders them particularly helpless against NK-mediated lysis. NK cells operate mainly via cytotoxicity: “natural” cytotoxicity caused by the secretion of perforin/granzyme granules inducing apoptosis, and the expression of factors such as FasL (ligand of Fas receptor, or CD95, tumor necrosis factor receptor superfamily member 6) or TRAIL (TNF-related apoptosis-inducing ligand), inducing apoptosis through the activation of death receptors; but also antibody-dependent cellular cytotoxicity (ADCC) mediated by the Fc receptor CD16 (FcyRIII), initiating their degranulation against antibody-coated cells. Moreover, NK cells produce many cytokines, most notably interferon gamma and TNF alpha. Because of these significant and fast lysis capabilities, in an antigen-independent manner and with no need for prior priming, many mechanisms exist to limit the risks of auto-reactivity. Indeed, NK activity is regulated by a complex integration of both activating and inhibitory signals—the sum of the interaction between the multitude of receptors on their surface with the different ligands on the target cells’ surface [2,3] (Figure 1). If the balance switches towards the inhibitory signals, the “healthy” cell is not destroyed. If the balance switches towards NK activation, the “stressed” cell, transformed by a virus or tumoral process, is destroyed. Most well-known interactions include the following: Firstly, ligands such as classical (e.g., HLA-C, HLA-Bw4) or non-classical MHC class I molecules (e.g., HLA-E) that are expressed on the target cell, and are specifically recognized by the NK inhibitory receptors (KIR or CD94/NKG2A), entailing a predominantly negative signal and, thus, avoiding autologous lysis through “self” recognition. Secondly, other ligands (e.g., MICA/MICB and ULBP) are induced by stress signals and expressed on the transformed cell, and are recognized by the specific activating NK receptors (e.g., NKG2D), entailing the activation signal and the destruction of the “non self”. Many other activating receptor ligands exists—mainly, NKp44, NKp46, NKp30, NKG2C, and KIR-S activators—but their nature and functions are still not completely understood, especially considering that certain ligands can be recognized at the same time by an inhibitory receptor and an activating one (e.g., HLA-E, recognized by the inhibitory CD94/NKG2A and the activating CD94/NKG2C receptors). For many years, researchers have wondered how the multitude of KIR receptors—both inhibitory and activating—can coexist in the same individual, with every single NK cell being tolerant towards self-HLA molecules [4]. This question was even more intriguing when KIR genes and HLA genes were segregated on different chromosomes. Finally, the description of “licensing” by Lanier et al. gave insight into the question—to be fully functional, an NK cell needs to express at least one inhibitory receptor specific to the “self” [5]. If an NK cells does not express any inhibitory receptor that recognizes the self-MHC class I (self), the cell is potentially auto-reactive, and is actually not fully functional, i.e., it is hyporesponsive to many stimuli and fails to reject MHC-class-I-deficient cells. There are two subpopulations of NK cells, defined by the intensity of the expression of their CD56: CD56^dim^ and CD56^bright^ NK cells. The CD56^dim^ subset has a less intense expression of the CD56 seen on flow cytometry. They represent approximately 90% of blood NK cells, are very powerfully cytotoxic, and mainly express CD16. CD56^bright^ NK cells (more intense expression of the CD56 in FC) represent 10% of circulating NK cells, but are predominantly present in the lymph nodes and some other organs. They are CD16^low^, and have less cytotoxic capability than their CD56^dim^ counterparts, but have a powerful proliferative potential. The phenotypic and functional profile of a mature NK cell is as follows: CD56^dim^, CD16^high^, KIR+, very cytotoxic. The immature NK cells have the following profile: CD56^bright^, CD16^low^, KIR^low^, NKG2A+, with a low cytotoxic but high proliferative capability. These differences in nature and functionality are important to potentiate the use and manipulation of NK cells in cell therapy.

## 2. Anti-Leukemic Effect of Allogenic NK Cells in Haploidentical Stem Cell Transplantation

Many hematological malignancies are not curable with chemotherapy alone, regardless of the dose of chemotherapy or irradiation used. Allogenic hematological stem cell transplantation (HSCT) is today the only curative option in a large number of hematological cancers. In allogenic HSCT, the main objective is to replace normal hematopoiesis, but also the immune system of the patient, with that from an allogenic donor. The donor’s immune system reaction against the receiver is called graft reaction. This reaction, abbreviated GvL for graft-versus-leukemia/lymphoma, is able to eliminate the residual malignant cells from the patient and control the hematological malignancy for an extended period of time. The major setback of this otherwise theoretically ideal solution is a hefty side effect: graft-versus-host disease (GVHD), where the donor’s immune system—essentially alloreactive T lymphocytes—attacks the host’s healthy tissue (notably the skin, the digestive tract, the liver, and the lungs, but any organ can potentially be impacted). In order to limit the risks of serious GVHD, most allogenic HSCTs are performed with HLA-identical donors and, hence, a complete HLA match between the donor and the receiver. However, since finding an HLA-identical donor is only possible in about 70% of all graft indications, alternative grafts, where one or many HLA mismatches exist, have been developed. In recent years, haploidentical familial hematopoietic stem cell transplants (HSCTs)—i.e., from a donor who shares one HLA haplotype with the receiver and not the other (for example, a father with his son, an uncle, a cousin, etc.)—have been used more often, accounting for up to 25% of all allogenic HSCTs in Europe [6]. In this particular situation, haploidentical NK cells from the donor can be activated. As a matter of fact, some class I HLA molecules (=KIR ligand) on the receiver’s leukemic cells’ surface are possibly not recognized by the inhibitory KIR receptors on the donor’s NK cells. This phenomenon is referred to as the GVL NK effect, mediated by the KIR/KIR-ligand mismatch. This notion has been reported in previous studies with a major anti-leukemic effect through this mismatch—especially in patients transplanted with a haploidentical donor in an AML setting [7,8]. Nevertheless, this anti-leukemic effect of the alloreactive NK cells after an allogenic HSCT is debatable, and some studies show contradictory results. Undoubtedly, NK cells, which possess important plasticity, adapt to their environment or present post-graft maturation delays, making them less effective and, thus, less cytotoxic [9,10]. In spite of this, even if the in vivo effect of the NK cells is not as well proven as T cells’ effect, the development of haploidentical HSCT since the late 1990s has help to shed light on this particular lymphocyte population, and to further exploit its potential for huge antitumor cytotoxicity [11,12].

## 3. CAR-T Immunotherapy

To date, allogenic HSCT is the oldest and most frequently used anti-leukemic immunotherapy. However, it certainly constitutes a radical (the receiver’s immune system is destroyed and replaced by the donor’s), non-specific, non-targeted immunotherapy, which explains its major toxicity—especially the GVHD risks, but also the infections related to the disruption caused to the patient’s immune system for the months or years after the transplantation. The development of chimeric antigen receptor (CAR) T cells has revolutionized cellular antitumor immunotherapy in an unprecedented manner [13,14,15,16,17,18,19]. A CAR-T cell is a regular T cell that has been genetically engineered to express a specific receptor of the targeted antigen. This chimeric receptor is composed of an extracellular single-chain variable fragment of an immunoglobulin (scFy) that is capable of specifically recognizing an antigen on the target cell, a transmembrane domain, and an intracellular domain used for signaling, composed of a cytoplasmic CD3ζ signaling domain that enables the CAR’s activation independent of pMHC (first-generation CAR), to which is infused a CD28 or a 4-1BB costimulatory domain to strengthen the mechanotransduction (second-generation CAR), or incorporating both (third-generation CAR) or single costimulatory domains associated with another transgene to elevate the expression cytokines in order to promote the CAR-T cell effector capability (fourth-generation). Fifth-generation CARs, currently being studied, have a truncated cytoplasmic IL-2 receptor β-chain domain with a binding site for STAT3/5, enabling full activation of the cell through three synergic signals (CD3ζ, costimulatory molecule, and JAK/STAT signaling) (Appendix A). The EMA has given approval to axicabtagene ciloleucel and tisagenlecleucel in R/R DLBCL [20], as well as tisagenlecleucel in young adults with B-ALL. The other autologous anti-CD19 or anti-BCMA CAR-T cells have early access in adult ALL, follicular NHL, and myeloma. With these innovative new therapies, we can observe 40–60% of long-term responses in patients who usually have a median overall survival of approximately 6 months with conventional salvage chemotherapy, completely changing the therapeutic landscape in these refractory–relapse hematological malignancies [21,22,23]. However, CAR-T therapy still faces many obstacles and practical issues for use [24]. CAR-T cells are developed from a patient’s own autologous cells, posing a problem regarding the quality of the T lymphocytes collected during lymphapherisis, along with the heterogeneity of the obtained products (e.g., effector capabilities, viability, expansion abilities, T senescence and exhaustion, etc.), essentially caused by the previous treatments, but also the patient’s immune system state. Moreover, these obstacles from the collection to the production process render the manufacturing procedure strenuous, long, and very expensive. Today, in the US, the cost of CAR-T cell therapy for a single patient with B-cell lymphoma averages USD 373,000. A new study by Prime Therapeutics using real-world data found that the total cost averages more than USD 700,000, and can exceed USD 1 million in some cases with post-CAR events. In France, where social insurance covers the whole cost of the CAR procedure, a CAR-T treatment’s price is approximately EUR 350,000, excluding the price of the hospitalization and adjacent treatments. The delay between the CAR-T cells’ indication and their infusion in the patient can reach 2 months, which is too long for a patient with a refractory uncontrolled hematological malignancy. Furthermore, the injected anti-CD19 CAR-T cells have their own toxicities—most notably inflammatory with the development of cytokine release syndrome (CRS), or neurological and hematological injuries (e.g., cytopenia, which can sometimes be persistent). Finally, loss of antigen target after CAR-T therapy is an important relapse cause, especially in CD19 B ALL. To palliate some of the obstacles linked to the autologous CAR-T setting, allogenic CAR-T cells have been developed with multiple advantages, such as their immediate availability, the standardization of the CAR product, the time to optimization, and even multiple modifications to increase the quality of T cells and decrease costs thanks to the industrialization of the process [25]. However, these lead to an increased risk of graft rejection (even failure) or GVHD. Others genetic modifications—such as TCR suppression or the introduction of a suicide gene—are necessary; some researchers even tried using double-negative CAR-T cells (CD4-CD8-) with no rejection or GVHD [26]. Some clinical studies are underway to test these allogenic CAR-T cells—mostly in B malignancies—with the usual fludarabine–cyclophosphamide (FluCy) lymphodepletion, to which alemtuzumab is added to decrease the risk of immune rejection. This procedure adds to the toxicity profile of the CAR, with the prolonged risk of immunosuppression and even long-term aplasia. In a recent trial by the UCART19 group, genome-edited, donor-derived allogeneic anti-CD19 CAR-T cells were given to adults and children with refractory B-ALL after FluCy conditioning with or without alemtuzumab. Only patients receiving alemtuzumab showed expansion and anti-leukemic activity, with a 67% CR rate at 28 days after infusion, but at the price of major toxicity. Indeed, cytokine release syndrome and neurotoxicity were present with a relatively higher incidence than what we see with autologous CAR therapy, along with GVHD, prolonged cytopenia, and 10% of treatment-related deaths, confirming high toxicity as the price to pay for efficacy [27]. Today, the most common use of allogenic CAR-T cells is as a bridging to allogenic HSCT—most notably in AML and myeloid neoplasms.

## 4. Immunotherapy with CAR-NK Cells

NK cells’ activity is completely independent of the antigen and of the HLA; they can be obtained from unrelated allogenic HLA-incompatible donors. This HLA incompatibility between the donor’s NK cells and the receiver’s tumor cells can promote antitumor alloreactivity of the NK cells via the KIR/KIR-ligand mismatch (see Section 1). Interestingly, and unlike allogenic T cells, NK cells do not cause any GVHD. The reasons for this phenomenon are still not completely understood, but activator receptors could hold the key. Healthy non-stressed (e.g., skin, digestive tract) host tissues do not present stress-induced ligands on their surface, sparing them from NK cytotoxicity. In HLA-incompatible murine HSCT models, donor NK cells even exert a protective effect against GVHD by destroying the receiver’s antigen-presenting cells. This effect on the receiver’s immune system could promote engraftment and make allogenic NK cells less susceptible to immune rejection. The possibility of off-the-shelf allogenic NK cells would present a great advantage in terms of time and cost [24,28,29] (Table 1). Even though CAR-NK cells present all of these advantages, this immunotherapy has some limits, explaining its delayed clinical use in comparison to CAR-T cells. First, their lifespan is relatively short, with a persistence of approximately 1–4 weeks. In clinical trials involving short-term (12–16 h) IL-2-activated CD3^−^CD56^+^ NK cells from familial haploidentical donors in patients with refractory AML, Miller et al. showed that lymphodepletion with fludarabine and cyclophosphamide is essential to NK cells’ expansion, and that the presence of IL-15 is a predictive factor for expansion and clinical response [30]. The in vivo expansion of NK cells also requires in vivo administration of IL-2, with significant toxic effects. The incorporation of cytokine transgenes such as IL-2 and IL-15 in the construction of CAR-NK cells has the potential to improve expansion and persistence [28]. It is hoped that IL-15 will stimulate T-regulatory cells significantly less than IL-2. Secondly, the limited number of NK cells in the bloodstream requires ex vivo expansion techniques to obtain the necessary doses after their harvest using apheresis. Different methods for ex vivo expansion and activation of NK cells have been investigated. These methods include several (usually 14) days of culture with cytokines—mainly IL15 and IL2. IL-2 is essential, and is currently used to promote NK cell expansion following allogeneic NK cell delivery, whereas IL-15 plays a crucial role in NK cell development and homeostasis. Large-scale expansion of NK cells has been also reported using feeder cells such as Jurkat T-lymphoblast or Epstein–Barr-virus-transformed lymphoblastoid cell lines (EBV-LCL). K562 is a leukemia cell line genetically modified to express membrane-bound forms of IL-15 and 4-1BB (CD137L). Recently, K562 cells have been engineered to express membrane-bound IL-21 along with CD137L, exhibiting some interesting results. A great benefit of these genetically modified K562 feeder cells is their potential use as frozen banks. Feeder cells are capable of inducing robust and sustained proliferation (from 100- to >10,000-fold expansion after 3 to 5 weeks in culture). Debate remains as to which method is best for achieving a higher expansion rate with limited exhaustion of NK cells and enhanced cytotoxic activity [31]. Moreover, the ability to achieve GMP-compliant protocols likely to support appropriate authorization and regulation is a major concern. Thirdly, NK cells have proven to be difficult to engineer, with high sensitivity to apoptosis and low levels of gene expression. Currently, the most successful alternatives for introducing genes into NK cells are either rapid transient expression by electroporation, or sustained but low expression by viral vectors. Thus, the optimization of transfection protocols remains an important element in the development of CAR-NK therapies [32]. Recently, improvements in lentiviral transduction of primary human natural killer cells in an ex vivo expansion setting have been made. Approaches with optimized methods for genetically engineering NK cells will promote the widespread application of CAR-NK cells [33].

Finally, the tumor microenvironment (TME) has a detrimental effect on NK cells, with a downmodulation of NK cellular metabolic activity in hematological malignancies, constituting a significant impediment in enhancing the antitumor effects of NK cells, and rendering them very dependent on the TME [34,35]. In the same way as IL-15 incorporation, the use of transgenes coding for chemokine receptors could promote CAR-NK trafficking in tumor sites. Combination with checkpoint inhibitors such as anti-PD1/anti-PDL1 could help to overcome the inhibitory effect of the TME and unleash the CAR-NK cells’ full potential. This effect seems to be successful in preclinical studies with murine models, but still needs proof in clinical studies with patients [36,37,38]. It is also important to note that the majority of data regarding checkpoints blocked with CAR cells are from CAR-T cell experience, with very little experience with CAR-NK cells.

## 5. CAR-NK Cells’ Sources

The favorable security profile, the relatively large proportion of NK cells in the lymphocyte population, and their antitumor cytotoxicity make NK cells one of the best candidates for allogenic adoptive cellular immunotherapy. A major advantage of allogeneic NK CARs is their ready availability. However, this requires the ability to freeze them, which is not yet developed for all NK cell sources. It would be then possible to generate a large homogeneous batch from a specific cellular source.

Many sources of NK cells have been used to produce allogenic CAR-NK cells.

### 5.1. NK-92 Cell Line

NK-92 is an interleukin-2-dependent, EBV-positive natural killer cell line derived from peripheral blood mononuclear cells from a 50-year-old white male with rapidly progressive non-Hodgkin lymphoma. They are highly cytotoxic. Their phenotype includes activating receptors NKp30, NKp46, and NKG2D. Interestingly, the only inhibitory receptors on their surface are represented by ILT2, CD94-NKG2A (recognizing HLA-E), and the inhibitory KIR receptor KIR2DL4 at a lesser level of expression. Thus, they have a very limited number of inhibitory KIRs that recognize class I HLA, HLA-C, or HLA-Bw4, which explains their very limited inhibition by the patients’ HLA molecules. However, they do not express CD16 on their surface, and cannot use any antibody-dependent cell-mediated cytotoxicity (ADCC). Their use can be very convenient because of the theoretical possibility to mass-produce and “infinitely” generate CAR-NK cells, with huge gains in terms of cell numbers, time, and cost. Unfortunately, because of their tumoral nature, these CAR-NK cells derived from the NK-92 cell line need to be irradiated before injection, altering their lifespan and division. Despite irradiation, they seem to preserve their cytotoxic capabilities, but their in vivo proliferation capabilities are sharply altered, with the disappearance of cells in vivo within 7 days of injections, necessitating multiple injections. Some evidence of the use of CAR-NK cells has been reported in human patients with CD33-CAR-NK cells derived from the NK-92 cell line in three patients with relapsed and refractory AML after lymphodepletion chemotherapy [39]. The three patients, treated with salvage chemotherapy, were then infused with the anti-CD33 CAR-NK cells at increasing doses (three escalating doses of irradiated CAR-NK-92 cells for each patient). Up to 5 × 10^9^ CD33-CAR-NK cells per patient were safely applied, with no substantial adverse effects or grade 3–4 toxicity; only transient fever was reported. However, even if the treatment was well tolerated and the safety of escalating infusions was established, the obtained response was very shortlived, with only 10 days for one patient and a maximum of 4 months for another one [39]. In vitro, the efficiency of the lentiviral CD33-CAR vector was well above 90% in NK-92 cells, but the cytotoxicity assay demonstrated a moderate enhancement of cytotoxicity against the human HL-60 promyelocytic leukemia cell line compared to parental NK-92 cells [39]. CAR-NK-92 cells have also been explored against multiple myeloma cells. Jiang et al. reported a strong antitumor activity of NK-92 cells transduced with anti-CD38 against myeloma cell lines and primary myeloma cells in vitro, as well as in a xenograft NOD-SCID mouse model [40]. Similar results were observed with NK-92 cells expressing an anti-CS1/SLAMF7 CAR [41].

In order to increase the efficiency of the CAR-NK cells derived from the NK-92 cell line, 4th-generation CARs were produced in a platform called UniCAR, initially used for T cells. The UniCAR system is composed of two elements: (i) the CAR-NKG2 directed against the peptide epitope E5B9—an antigen that is not naturally expressed on the cells’ surface; and (ii) a bispecific component named target module (TM). This bispecific module expresses the E5B9 antigen on one side and an antibody specific to a tumoral antigen of the other (Figure 2). This construct enables a contact between the CAR-NKG2 cells (via the E5B9 antigen) and the target cell (via the specific antibody). In addition to this, the effect can be immediately stopped (on/off effect), because the lifespan of the TM is very short. Once the steady (continuous) infusion of the TM stops, the TM is eliminated and UniCAR is inactivated. Indeed, this system enables the targeting of a large selection of tumoral antibodies by changing the TMs that can be given simultaneously or in a sequential manner [42].

### 5.2. NK Cells Derived from the Umbilical Cord

Umbilical cord blood is a rich and interesting source of NK cells, because the international blood banks have hundreds of thousands of umbilical cord blood (UCB) units available for use. NK cells represent a fair 15 to 30% of umbilical cord lymphocytes. Their phenotype is more immature than that of peripheral blood cells found in adults, with a very particular profile composed of low CD16 (low ADCC capacity), CD56 bright (bestowing a more proliferative asset than a cytotoxic one), NKG2A+ (an inhibitory receptor that recognizes HLA-E), and low KIR (less functional than their counterparts) [43]. However, these limitations can be overcome by ex vivo expansion and activation using cytokines and transfection, and in our experience, CAR-NK cells from cord blood have the same activation profile in flow cytometry in comparison with CAR-NK from blood cells. Even if the NK cells in the UCB are more or less scarce, they have a very important proliferation capability, and are very sensitive to cytokinic stimulation, enabling their ex vivo expansion. A team from MD Anderson directed by Tezvani et al. recently published a very encouraging phase I/II clinical trial testing the injection of CAR-NK-CD19 cells derived from fresh UCB in 11 patients suffering from relapsed refractory B-cell malignancies raging from non-Hodgkin lymphomas to chronic lymphocytic leukemia (ClinicalTrials.gov, NCT03056339). The NK cells were transduced with a retroviral vector expressing genes coding for the anti-CD19 CAR, IL-15, and a suicide caspase-9 gene (Figure 3). The modified NK cells were infused in a single perfusion with one of the three doses used in the study (1 × 10^5^, 1 × 10^6^, or 1 × 10^7^ cells per kilogram) after the lymphodepletion chemotherapy (fludarabine and cyclophosphamide). HLA incompatibilities between the UCB NK cells and the receivers (patients) were numerous. Initially, the authors tried to find umbilical cord units presenting HLA types promoting NK alloreactivity, but did not subsequently take the HLA type into account. Remarkably, no adverse effect or toxicity was noted, especially regarding CRS or neurotoxicity, with the maximal tolerated dose not reached in the trial. Responses promptly appeared in the first month after injection, and were profound, with a complete remission in 7 patients out of the 11 tested, and a return to the CLL state in a Richter transformation. Nonetheless, it is important to showcase the fact that half the patients had received maintenance therapy after the NK cells’ injection, consisting of lenalidomide, rituximab, venetoclax, or HSCT. NK cells were detected up to 12 months after the injection. Most likely, the adjunction of the IL-15 gene in the CAR construction contributed to the prolonged persistence of the CAR-NK cells, even though the mechanisms of persistence have not yet been elucidated. Moreover, a possible and very important technical setback is the production of the CAR-NK cells directly from fresh UCB, along with the immediate injection after production, without any congelation step. Indeed, the use of cryopreserved cord blood from international banks to produce CAR-NK cells should probably not be recommended, due to the low efficiency of NK cell selection from cryopreserved cells. The expansion from fresh cord blood as in the experience of the MD Anderson team could be an option—or rather, the establishment of banks of selected cryopreserved NK cells that can be thawed and expanded for transduction. It should also be noted that cultured, activated NK cells—and more specifically, CAR-NK cells—demonstrate a low viability after cryopreservation, and that in current technical conditions, CAR-NK cells should be administered immediately after culture. Even if the specific long-term efficacy of the CAR-NK cells is difficult to evaluate (because of the maintenance after injection) and the feasibility of the process on frozen cells is yet to be determined, the results are very promising [44].

### 5.3. NK Cells Derived from Peripheral Blood

As previously indicated, most studies with adoptive NK therapy (excluding CAR-NK cells) have used cells derived from haploidentical related or unrelated donors, and have shown an antitumor response in pediatric and adult AML [29,46]. Lymphocytes were obtained from lymphapherisis and then enriched with NK cells after depletion of the T and B lymphocytes (CD3+ and CD19+), and positive selection of the CD56+. cells was activated overnight with IL-2 and then re-injected in the patients [47,48]. These cells represent a major benefit thanks to their mature, well-differentiated, and mostly cytotoxic CD56dim phenotype. The drawback, however, is the need to collect the cells from healthy adult volunteers, making the process somewhat more complicated than using UCB. Additionally, adult NK cells have less proliferative ability than cells derived from UCB. Nevertheless, similar protocols could be applied to the expansion of NK cells issued from CB or PB, and in our experience, some simple, GMP-compliant culture conditions in the presence of cytokines can enhance their expansion by 25–55-fold, allowing their easy application in CAR-NK therapies [49]. Finally, if NK cells from PB are highly cytotoxic compared with CB NK cells, it remains to be demonstrated whether the choice of the donors in term of HLA incompatibility should be different in the context of CAR-NK therapies.

Anti-CD19 CAR-modified memory-like NK cells have been generated from autologous PBMCs in patients suffering from DLBCL [50]. Cytokine-induced memory-like NK cells were first described in a murine model by activating NK cells with IL-12, IL-15, and IL-18, amplifying their antitumor response, with a significant decrease in tumor burden and enhanced survival. The identification of IL-12-, IL-15-, and IL-18-induced memory-like NK cells in humans would allow their use in cellular antitumor immunotherapy. Human memory-like NK cells have enhanced proliferative abilities, expressing the high-affinity IL-2 receptor αβγ (IL-2Rαβγ) and an increased production of IFN-γ after cytokinic “re-stimulation” or via activator receptors [51,52]. The anti-CD19 CAR-modified memory-like NK cells presented high cytotoxic activity ex vivo against primary lymphoma cells derived from patients with B malignancies and in a xenograft Raji + murine model, showcasing the importance of this achievable and interesting approach in an autologous setting.

Another approach that can be used and derived from PBMCs was described by the team of Leivas et al. [53]. Effector cells were chosen between activated and expanded NK cells (NKAE) and memory T cells (CD45RA- T cells). NK or T cells were transduced after culture with an NKG2D-4-1BB-CD3z-CAR vector, using the NKG2D activator receptor expressed on both NK and T cells, and 4-1BB for transduction (see above). NKG2D has specific ligands including MICA, MICB, and ULBP, which are expressed on stressed cells and on many tumoral cells—notably in multiple myeloma. In vitro, transduction of memory T cells was more stable than that of their NK counterparts. However, CAR-NKAE cells exhibited greater in vitro cytotoxicity against MM cells, while protecting healthy cells from damage. Furthermore, in their murine xenograft myeloma model, CAR-NKAE cells displayed highly efficient cytotoxicity and enhanced anti-myeloma activity in comparison to their T counterparts, demonstrating the feasibility of this strategy.

Another approach is a CRISPR-targeted CAR gene insertion using Cas9/RNP and adeno-associated virus gene delivery. Using this technique, a CD33-targeting CAR-NK with different transmembrane and signaling domains (CD4/4-1BB+CD3ζ and NKG2D/2B4+CD3ζ) was generated, and seemed to show enhanced anti-AML activity in primary NK cells [54].

### 5.4. NK Cells Derived from iPSCs

Induced pluripotent stem cells (iPSCs) are differentiated cells that are genetically reprogrammed into becoming pluripotent stem cells. The reprogramming process of a mature differentiated cell into an iPSC consists of genetically modifying the cell and making it express four genes that bestow it with the same characteristics as an embryonic stem cell, without the need to use a human supernumerary embryo: *Klf4*, *C-myc*, *Oct3/4*, and *Sox2*. Thus, the differentiated adult reprogrammed cell gains the potential of unlimited proliferation, and can differentiate into any cellular type after the adjunction of the specific growth factor, which renders it particularly interesting for use in many medical domains—especially regenerative medicine, cellular therapies, or modularization of genetic disease. This discovery was rewarded with the 2012 Nobel Prize in Medicine to Shinya Yamanaka—a researcher at the University of Kobe (Japan) [55]. Multiples sources for differentiated cells can be used, including blood cells, fibroblasts, epithelial cells, mesenchymal cells, etc. Due to their immortality, a single iPSC is sufficient to produce and generate a universal CAR-NK “off the shelf” product that is ready for use. NK cells generated from iPSCs have an immature phenotype with low CD16, high NKG2A, and low KIR, conferring poor cytotoxicity but very important proliferation capabilities. To enhance their cytotoxicity and enable their persistence, it is possible to add genes for the expression of CD16 and IL-15 or IL-2 [56,57].

The FT596 product is the first “ready to use”, universal, and allogenic CAR NK cellular product derived from iPSCs to be authorized for use in clinical studies in the USA [58]. It is composed of an anti-CD19 CAR optimized for NK cells, with a transmembrane domain for the activator receptor NKG2D, a 2B4 costimulatory domain, and a CD3ζ signalization domain. Two key components have been added: (i) a novel, high-affinity, non-cleavable CD16 Fc receptor (hnCD16) that enables tumor targeting and enhanced antibody-dependent cell cytotoxicity without negative regulation, in combination with a therapeutic monoclonal antibody targeting the tumor cells; and (ii) an IL-15/IL-15 receptor fusion protein (IL-15RF) promoting cytokine-independent persistence (Figure 4). When used in combination with monoclonal antibodies such as rituximab (anti-CD20), the hnCD16 Fc receptor of FT596 binds to the Fc portion of the monoclonal antibodies covering the tumoral cells, activating the NK cells, the secretion of cytokines, and enhanced ADCC. IL-15RF promotes the cytotoxicity of the NK cells and the activated antitumor T cells. FT596′s action uses three different and complementary pathways: the anti-CD19 CAR, ADCC via the anti-CD20, and IL-15. A communication during ASH 2020 reported the case of a 76-year-old woman with relapsed refractory DLBCL after eight lines of treatment, including ASCT, autologous adoptive T-cell therapy, and activated haploidentical NK cells [59].

The patient received a unique dose of 30 × 10^6^ FT596 cells after lymphodepletion monotherapy. No toxicity of the FT596 cells was reported except for neutropenia, which was spontaneously resolved. The evaluation of the tumoral response at the one-month threshold showed a partial response based on the 2014 Lugano criteria, with a decrease of more than 70% in the ^18^F-Glu absorption and a reduction of more than 50% in the tumoral size. A second dose of FT596 is underway for this patient. A phase I study is also underway, with an estimated enrollment of 285 patients (clinicaltrials.gov: NCT04245722).

Key points can be found as Appendix A. Abbreviations can also be found in Appendix A.

## 6. Conclusions

Allogenic CAR-NK cells offer many interesting perspectives that can answer some problems faced in the use of autologous CAR-T cells. The major setback facing allogenic NK cells to date has been their relatively short persistence and the lack of standardized and efficient transduction protocols. However, recent constructions associating CAR and IL-15 or stable CD16 with enhanced ADCC seem very promising. CAR-NK therapy is still largely underdeveloped in comparison to CAR-T cells; as of the 1 May 2022, there were 35 trials involving CAR-NK cells (Appendix A) versus 1181 trials involving CAR-T cell therapy on clinicaltrials.com, confirming the continued scarcity of data on this innovative therapy. These clinical trials include CAR-NK cells targeting B-NHL (CD19, CD22), AML (anti-CD33, anti-CLL1, anti-NKG2D), CD7+ hemopathies (anti-CD7), multiple myeloma (anti-BCMA), epithelial ovarian cancer (anti-mesothelin), castration-resistant prostate cancer (anti-PSMA), pancreatic cancer (ROBO1 CAR NK), metastatic solid tumors (anti-5T4, anti-MUC1, anti-NKG2D), and glioblastoma (anti-HER2). Through its numerous assets—especially the ability to constitute an allogeneic bank with less toxicity—CAR-NK therapy is a promising prospect in the fight against cancer.

## Figures and Tables

**Figure 1 cancers-14-03839-f001:**
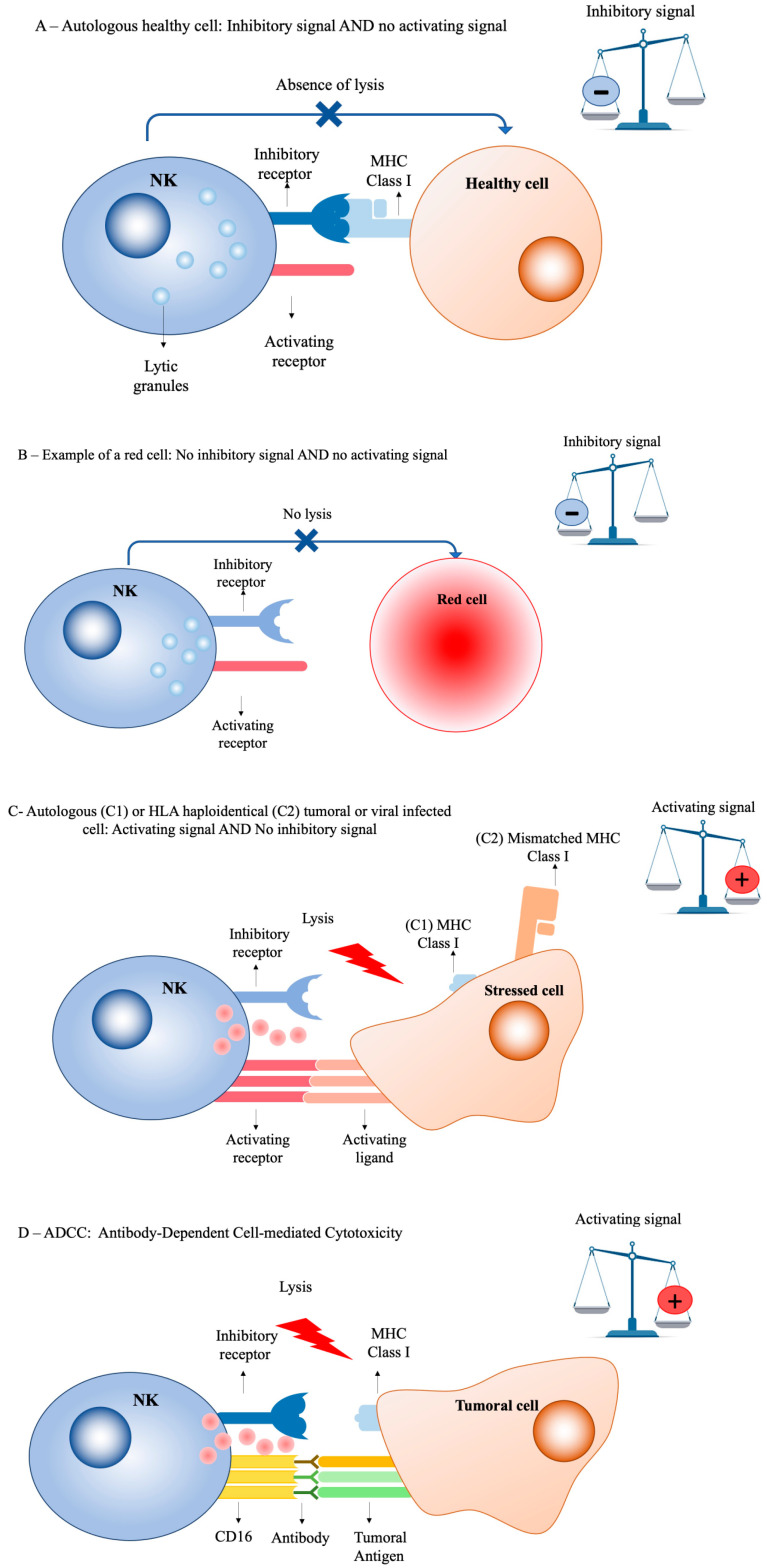
**NK cells’ cytotoxicity: A balance between activation and inhibition**. (**A**) In autologous settings, NK inhibitory receptors (e.g., KIR or CD94/NKG2A) recognize self-HLA class I ligands. Autologous healthy cells do not express any stress ligand; inhibitory signals dominate. (**B**) Red cells do not express HLA class I molecules or stress ligands; inhibitory signals dominate. (**C**) (C1): Stressed cells (tumor or virus-infected cells) can negatively modulate HLA class I ligands and express activating stress ligands. (C2): Haploidentical recipient leukemic cells express activating stress ligands and HLA class I molecules that are not recognized by donor NK cells. Alloreactive donor NK cells kill recipient tumoral cells via a KIR ligand mismatch. (**D**) NK cells express Fcγ receptors such as CD16. CD16 binds to the Fc region of the antibody, which binds to tumoral antigens and provoke NK cell degranulation.

**Figure 2 cancers-14-03839-f002:**
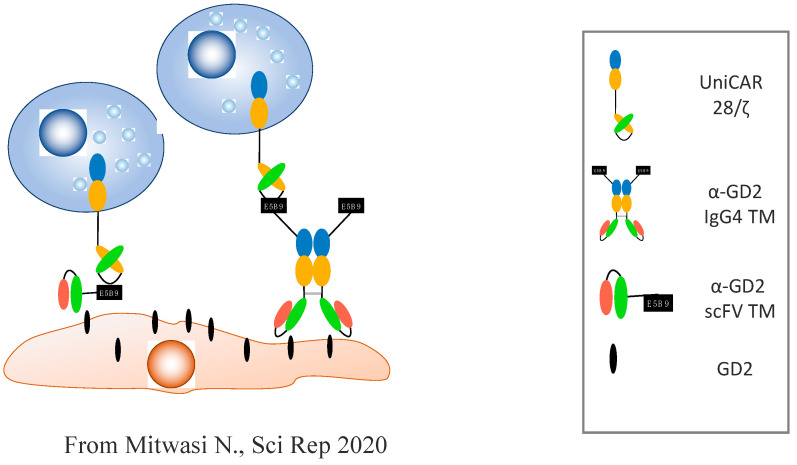
**UniCAR-NK-92 action against GD2+ tumoral cells:** UniCAR NK-92 is derived from the human tumoral line NK-92. It is composed of (i) an extracellular antibody with a single-chain variable fragment (scFv) directed against the peptidic epitope E5B9, which is not expressed on the host’s cells; (ii) the costimulatory domain CD28; and (iii) the signalization fragment CD3ζ. NK-92 cells modified to express UniCAR can be redirected against the tumor cells expressing the antigenic target GD2 via specific “Target Modules”. These TMs are composed of a bispecific antibody that recognizes the GD2 antigen on the tumoral cell’s surface on one side, and the E5B9 epitope that interacts with the NK-92 UniCAR on the other. Adapted from Ref. [42].

**Figure 3 cancers-14-03839-f003:**
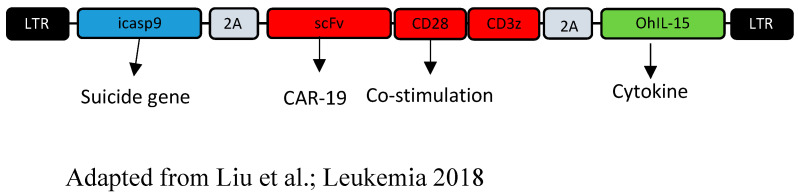
**CAR-NK cells derived from umbilical cord blood:** The CAR-NK cells’ construction, described by Rezvani et al. [45] from the MD Anderson Center, was achieved from a classical CD19 CAR with a costimulatory CD28 domain, to which a suicide gene and an IL-15 gene were added.

**Figure 4 cancers-14-03839-f004:**
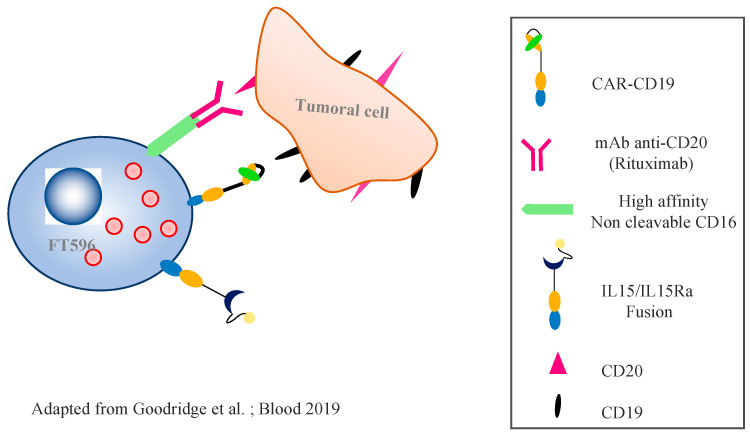
**Mechanism of action of the FT596 cell, derived from iPSCs:** The FT596 cell is composed of an anti-CD19 CR optimized for use in NK cells, with a transmembrane domain for the activating receptor NKG2D, a 2B4 costimulatory domain, and a CD3ζ signalization domain. A high-affinity, non-cleavable CD16 Fc receptor (hnCD16) was added to increase antibody-dependent cell cytotoxicity by preventing negative regulation of the CD16, and by enhancing the CD16 bond with the tumor cells. A fusion IL-15/IL-15 receptor (IL-15RF) promoting cytokine-independent persistence was also added. When used in combination with monoclonal anti-CD20 antibodies (e.g., rituximab or obinutuzumab), the hnCD16 Fc receptor of FT596 binds to the Fc portion of the monoclonal antibodies covering the tumor cells, activating the NK cells, the secretion of cytokines, and enhanced ADCC. IL-15RF promotes the cytotoxicity of the NK cells and the activated antitumor T cells. FT596′s action uses three pathways: the anti-CD19 CAR, ADCC via the anti-CD20, and IL-15 [58].

**Table 1 cancers-14-03839-t001:** Advantages and disadvantages of autologous CAR-T cells and allogeneic CAR-NK cells.

Autologous CAR-T		Allogeneic CAR-NK	
Advantage	Disadvantage	Advantage	Disadvantage
Better expansion	Preparation time	“off the shelf” and immediate availability	Short lifespan
Better persistence	Higher cost; 1 patient = 1 product	Lower cost; 1 source = many products	Difficult to modify, especially with frozen cells
Autologous setting (the patient is their own donor)	Variable cell quality	Multiple allogeneic sources (PBMC, UCB units, NK92 cell line, iPSC)	If derived from immature cells (UCB/iPSC) with a CD56 bright, KIR neg, NKG2A+, and low CD16, meaning proliferative but less cytotoxic and, thus, the need for further manipulation; if derived from tumor cell lines, need to be irradiated
Easier modification	Toxicity (CRS, neurotoxicity, persistent cytopenia)	No GVHD, good safety	If derived from PBMCs (CD56dim, CD16+, KIR+), a need for a specific haploidentical donor and promotion of proliferation. The criteria of choice of the donor are to be defined (interest of the mismatch KIR ligand?)
	T-cell senescence (high tumor burden, consistent antigen stimulation)	Superior quality and product homogeneity	Inhibition by inhibitory receptors and the need of further modification to surpass it (HLA-based donor or addition of a cytokine/cytokine receptor fusion protein)
	Distinct manufacturing methods with heterogeneous products dependent on the cells and the collection, the patient	Possible expansion to produce many batches from a single source	Risk of immune rejection
Proven clinical efficacy in B-cell hemopathies and multiple myeloma		Three mechanisms of action: classical cytotoxicity via the CAR and ADCC (with the possibility to target a second antigen, limiting the risk of relapse by antigen loss), cytokine pathways (IL-15)	Preclinical observations

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
