# Peer review of "CAR-NK Cells: A Chimeric Hope or a Promising Therapy?"

_cancers, 2022, doi:10.3390/cancers14153839_

Round 1

Reviewer 1 Report

Main message:

In this review Sabbah et al. provide an overview on the history and recent advances and disadvantages of CAR-NK cell therapy in comparison to conventional CAR-T cell therapy for the treatment of hematological malignancies. Herein evidence is presented, that applying the CAR technology to other  immune cells besides T cells, in particular innate immune cells like NK cells may translate into also feasible, safe, effective, and affordable therapies for future clinical applications. The review discusses different NK cell sources, optimized CAR-constructs, CAR-engineering techonolgies, expansion protocols resulting in multiple anti-tumor capacities besides the CAR-medieted toxicity and low alloreactive potential considering NK cells as ideal candidates for CAR-immunotherapy.

The review is interesting, well-structured and comprehensive and contains important aspects and landmark papers that should be considered for CAR-NK-based therapy against hematological malignancies.

Minor comments:

The sentences: “Today, the most common use of allogenic CAR-T cells is a bridging to allogeneic HSCT” should be discussed. CAR-T therapy in general and allogeneic HSCT both are considered (alternative) treatment approaches for hematological malignancies.

The manuscript should be edited for formatting issues.

Author Response

Thank you for your very insightful review and we are glad you have liked the article and found it interesting. We tried to make a complete and easy to read review with recent data. 

Regarding the allogenic CAR-T cells as a bridge to HSCT, we meant that most studies with allogenic CAR-T cells, especially in AML, use it as a bridge, because of common antigens with hematopoietic cells thus the need to back it up with HSCT. In other indications, they are both alternative treatments approaches. Changes have been made to make it clearer. 

Formatting issues fixed and language related changes also.

Thank you again for your review. 

Reviewer 2 Report

The present manuscript addresses the recent advancements made in the field of CAR-NK cells. The manuscript is well-written and covers important aspects of CAR NK cells. A more structured revised version with additional background information in a language accessible to non-specialist readers would substantially improve the value and impact of this article. Any revision of the manuscript should address the comments listed below.

Comments:

1.    Authors should revise the abstract for fluency.

2.  The authors need to add more in-depth information regarding the mechanism of CAR NK activity and its clinical applications against various cancers.  

3.    Describe how CAR NK cells could be used to develop innovative anti-cancer therapies for different human cancers.

4.  For the ease of readers and a better understanding of CAR T cells, the authors should add a figure showing different generations of CAR T cells.

5.   Please add the references for every scientific statement mentioned in the review article. For example, Line 78-88 and 153-155 should have a reference at the end.

6.    Authors should also discuss the therapeutic advantages of combination therapies such as immune checkpoint blockade for improving the efficacy of CAR NK cells.

7.   Please add the relevant research article instead of referring to the review article wherever it is required as it takes the credit from the actual scientific addition.

8.    In Fig.1, the figure legends are too small to read.

Author Response

Thank you for your valuable review. We have revised the article and included the relative information to your comments.

  1. Abstract has been rephrased and checked for fluency.
  2. We discussed the functionality of NK cells and their activation/inhibition dependant on their receptors. The CAR work via the targeting of a TAA (tumor associated antigen) and its activation. We added information on the mechanism of action dependant on each source.
  3. We added the applications in different haematological malignancies and in different solid cancers.
  4. figure added as supplementary.
  5. references added.
  6. TME and immune checkpoint paragraph added explaining the possibility of ue of checkpoint blockade with some preclinical data in CAR NK cells and CAR T cells but very few in clinical studies.
  7. References adjusted when necessary.
  8. Figure 1 modified for easier reading.

Thank you again for your review.